# Subchorionic Hematoma Association with Pregnancy Complications and Outcomes in the Third Trimester

**DOI:** 10.3390/jpm13030479

**Published:** 2023-03-07

**Authors:** Haixia Huang, Huan Han, Han Xie, Hao Ying, Yirong Bao

**Affiliations:** 1Shanghai Key Laboratory of Maternal Fetal Medicine, Shanghai 200092, China; 2Department of Obstetrics, Shanghai First Maternity and Infant Hospital, School of Medicine, Tongji University, Shanghai 200092, China

**Keywords:** subchorionic hematoma, third trimester, pregnancy complications and outcomes

## Abstract

Introduction: Our objective was to explore the clinical features, pregnancy complications, and outcomes of subchorionic hematomas (SCHs) in the third trimester. Material and methods: This was a retrospective analysis and evaluation of 1112 cases diagnosed with SCHs from January 2014 to December 2020. Comparisons were performed according to the clinical features (e.g., number of pregnancies, parity, gestational weeks, and age), pregnancy complications, and outcomes associated with SCHs. Results: In total, 71.85% (799/1112) of the patients were diagnosed with different pregnancy complications. The overall rates of gestational diabetes mellitus (GDM), hypertensive disorder complicating pregnancy (HDCP), premature rupture of membranes (PROM), and IVF were 12.14%, 7.55%, 17.27%, and 10.34%, respectively. The positive rates for newborn outcomes such as premature birth and low birth weight (LBW) were 9.35% and 6.47%, respectively. There was a significant relationship between repeated pregnancies and the incidence of GDM (*p* < 0.05), but not HDCP, PROM, or IVF. The proportion of SCH patients who conceived through IVF was significantly higher among primiparas than among multiparas (*p* < 0.05), but was not significantly different in terms of GDM, HDCP, or PROM. Premature birth was not a high-risk factor for most SCH patients with HDCP, IVF, or PROM (*p* < 0.05), most of whom delivered at term. The rate of cesarean sections for SCH patients with GDM, HDCP, or IVF was significantly higher than that for vaginal deliveries (*p* < 0.05), but this was not affected by age. Conclusions: The coexistence of SCHs with HDCP, IVF, or PROM lacked an effective predictive value for premature birth, but increased the rate of a cesarean section.

## 1. Introduction

A subchorionic hematoma (SCH) is defined as a crescent-shaped, echo-free area between the chorionic membrane and the myometrium. It is the most common sonographic abnormality during early pregnancy and the most common cause of first-trimester bleeding. The incidence of an SCH in pregnancy is 0.46–39.5% [1,2], depending on the population studied, SCH definition, and gestational age at diagnosis.

Although the clinical significance of SCHs remains controversial, recent publications have suggested that the frequency of SCHs has significantly increased with an increasing use of in vitro fertilization (IVF) techniques [3]. SCHs have been reported to have diverse outcomes in non-IVF pregnancies, including an increased risk of pregnancy loss, abruption, preterm premature rupture of membranes (PPROM), and fetal growth restriction [4,5,6]. Recent studies have suggested that first-trimester SCHs are associated with both a shorter cervical length and preterm birth [7]. Additionally, pregnant women with SCHs in the first trimester show changes in vaginal flora in the second trimester, which suggests a possible association between SCHs and vaginal flora changes [8]. However, scientific research has reported that SCHs do not represent a risk factor for complications of pregnancy [9,10,11], concluding that SCHs before 14 weeks of gestation are not associated with either pregnancy loss before 20 weeks of gestation or diverse pregnancy outcomes after 20 weeks of gestation [12,13].

In previous studies, the effect of hematomas on pregnancy complications and outcomes in SCH patients in the first and second trimesters was evaluated. However, there is still insufficient evidence to evaluate the risk of pregnancy complications for patients diagnosed with SCHs in the third trimester.

## 2. Materials and Methods

### 2.1. Study Population

This was a retrospective study conducted at Shanghai First Maternal and Infant Health Hospital, China, from January 2014 to December 2020. The study evaluated 1112 pregnant patients with accompanying SCHs, 799 of whom were diagnosed with different pregnancy complications in the third trimester (gestation ≥ 28 weeks). The criterion for inclusion in this study was the presence of SCHs, diagnosed by ultrasound (by sonographers with a clinical practitioner certificate in the scope of medical imaging and radiotherapy) throughout pregnancy, and the exclusion criterion was patients who delivered before 28 weeks. The study was approved by the ethical committee.

### 2.2. Data Analysis

Data on the background characteristics of the patients, including gravidity, parity, age, and in vitro fertilization (IVF) use, and the main pregnancy outcomes such as gestational age, mode of delivery, and the occurrence of premature rupture of membranes (PROM) were collected. We also analyzed the main complications of pregnancy—namely, gestational diabetes mellitus (GDM, diagnosed according to the International Association of Diabetes and Pregnancy Study Groups criteria) and hypertensive disorder complicating pregnancy (HDCP, diagnosed according to the American College of Obstetricians and Gynecologists clinical guideline)—and the following complications of delivery: premature birth and low birth weight (LBW). All pregnancy-related diseases were diagnosed in strict accordance with the updated clinical guidelines and *Obstetrics and Gynecology* textbook.

### 2.3. Statistical Analyses

The statistical analysis was performed using IBM SPSS Statistics 20 (SPSS Inc., Chicago, IL, USA). Group comparisons with respect to categorical variables were performed using chi-squared tests, and the t-test (normally distributed data) or one-way ANOVA for analyzing continuous variables were applied. A probability value of *p* < 0.05 was considered to be statistically significant.

## 3. Results

From the overall data, we were able to collect data on the background characteristics of the patients, including gravidity, parity, age, gestational age at delivery, and mode of delivery. In total, 71.85% (799/1112) of the patients were diagnosed with one or more pregnancy complications in the third trimester (gestation ≥ 28 weeks), as follows: GDM (12.14%, 135/1112); HDCP (7.55%, 84/1112); fetal growth restriction (1.08%, 12/1112); cervical incompetence (0.72%, 8/1112); systemic immune disease (1.98%, 22/1112); and intrahepatic cholestasis of pregnancy (1.80%, 20/1112). The positive rates of PROM and IVF were 17.27% (192/1112) and 10.34% (115/1112), respectively, and the rates of premature birth and LBW were 9.35% (105/1112) and 6.47% (72/1112), respectively. The mean maternal age was 31.10 ± 3.73 years (19–49 years), and the mean gestational age was 38.6 ± 2.00 weeks (28.1–41.4 weeks).

On the basis of the statistical results (Table 1), there was a significant relationship between gravidity of two or more and the incidence of GDM (*p* = 0.001), but not HDCP, PROM, or IVF (*p* > 0.05). Among pregnancies conceived through IVF, the SCH proportion was significantly higher for primiparas than for multiparas (*p* < 0.05), but no significant difference was found for the incidence of GDM, HDCP, or PROM. Although there was a significant difference between the full-term group (delivered after 37 gestational weeks) and premature group (delivered between 28 and 36.6 gestational weeks) related to the presence of HDCP, IVF, or PROM (*p* < 0.05), the percentage was much higher in the full-term group. Thus, premature birth was not a high-risk factor for most SCH patients with HDCP, IVF, or PROM. The rate of cesarean sections was significantly higher than that of vaginal delivery among SCH patients with GDM, HDCP, or IVF (*p* < 0.05), but the cesarean section rate was independent of age.

Among our subjects (Table 2), significant differences were found between gestational weeks and the incidences of HDCP, IVF, and PROM for patients suffering from SCHs (*p* < 0.05), but premature labor (limited to less than 37 weeks) was not a high-risk factor in the groups. This finding requires further study, with an expansion of the sample size and an analysis of the received data in more detail. For instance, future studies will be carried out to investigate the relationships between pregnancy features before 28 gestational weeks and pregnancy outcomes and between premature birth and the complications of cervical incompetence as well as pregnancy-associated infections and systemic immune diseases, among others. As shown in Table 1, age was not a significant factor affecting complications such as HDCP and PROM, or IVF.

In our further analysis of the effect of gravidity (the frequency of pregnancy) and parity on pregnancy weeks (Table 3), a statistical analysis using one-way ANOVA showed that different gravidities and parities had an effect on gestational weeks at delivery. However, the mean number of gestational weeks at delivery in all groups was equivalent to full term, which suggested that premature labor was not a high-risk factor for gravidity or parity of two or more, as shown in Table 1 and Table 2.

## 4. Discussion

The finding of an SCH in a pregnant woman on a first-trimester ultrasound examination is fairly common, but quite worrisome [14]. Intrauterine hematomas vary in size and location in the first trimester, but are mostly smaller than those that occur in the second or third trimester [15]. Several studies have shown that in ongoing spontaneous pregnancies, the presence of intrauterine hematomas is associated with an increased risk of adverse pregnancy outcomes, including pre-eclampsia, miscarriage, fetal growth restriction, and preterm delivery [6,16]. The published literature contains a variety of conclusions for characterizing the relationships between SCHs and different pregnancy outcomes, which can make it difficult to follow the progression of SCHs in a pregnancy or to compare SCHs in different pregnancies. This is the reason why providing good counseling to patients on the prognostic importance of their SCHs is difficult.

The major goal of our study was to determine the effect of SCHs on pregnancy outcomes from our single-center study, which was described in the Results section of this paper. We did not find a correlation between the presence of SCHs and late-pregnancy complications, including HDCP, preterm labor, PROM, GDM, or fetal growth retardation. Although this single-center study was limited and small, it plays an important guiding role in hospital diagnosis and treatment. A secondary goal was to determine whether the clinical characteristics of SCHs affected pregnancy outcomes by collecting database resources reported by diverse clinical research centers. We reviewed the systematic search terms “subchorionic/hematoma” from the computerized databases Web of Science and PubMed, covering the period from January 1976 to December 2021, and focused on the pregnancy complications and outcomes associated with SCHs.

### 4.1. Risk Factors for SCHs and Related Pregnancy Complications

At present, no single factor can explain the etiology and pathogenesis of all SCHs. According to the literature reports, the high-risk factors for SCHs and the relationship with diverse pregnancy outcomes are as described below.

#### 4.1.1. Clinical Features, Location, and Duration of Hematomas

A retrospective study of the clinical features of patients with intrauterine hematomas in the second and third trimesters evaluated the risk factors for poor pregnancy outcomes among 398 patients. The results revealed that intrauterine hematomas in the second and third trimesters were a sign of pathological pregnancy, resulting in adverse outcomes, and that maternal age, gestational age at first diagnosis, location of the hematoma, and accompanying contraction were risk factors for poor pregnancy outcomes [15]. In addition, although the location and duration of the intrauterine hematoma had a strong prognostic value, the volume of the intrauterine hematoma, gestational age at diagnosis, and the prognostic value of vaginal bleeding remained controversial. This was reported by Xiang et al., who performed a literature review with search terms that included intrauterine/subchorionic/retroplacental/subplacental/hematoma/hemorrhage/bleeding/collection/fluid, covering the period from January 1981 to January 2014 [2].

#### 4.1.2. Assisted Reproductive Technology

A retrospective case-control study was conducted to compare the obstetric and perinatal outcomes of 350 pregnancies with intrauterine subchorionic hematomas and 350 matched controls, and found that the incidence of intrauterine subchorionic hematomas among pregnant women who conceived by in vitro fertilization and embryo transfer (IVF-ET) was 13.5% in the first trimester [17]. Researchers from another group analyzed 1097 pregnancies achieved by IVF-ET or frozen hawed embryo transfer (FET) in a retrospective cohort study, and found that SCHs were associated with a lower birth weight in singleton pregnancy; however, an SCH did not increase the pregnancy loss rate in IVF/ICSI patients [18]. In addition, to determine the correlation between subchorionic hematomas and IVF pregnancies, a study analyzed 194 pregnancies (67 by IVF and 127 by non-IVF approaches) achieved by infertility treatments and revealed that the frequency of SCHs was high in IVF pregnancies and FET (parity ≥ 1), and concluded that a blastocyst transfer may contribute to SCH onset (16.6% vs. 5.1%; *p* < 0.01) [19]. This conclusion has been unanimously accepted by scientists from different institutions [18,19].

#### 4.1.3. Gestational Hypertension

The impact of SCHs on gestational hypertension is controversial. A retrospective study of 185 pregnant women hospitalized due to symptoms of a threatening miscarriage (119 women with SCHs and 66 patients without SCHs) found no significant correlations between SCHs or vaginal bleeding and preterm labor, intrauterine growth retardation, pregnancy-induced hypertension syndrome, abnormal volume of the amniotic fluid, parity, and order of gestation and delivery [20]. However, scholars have reported that the incidences of gestational hypertension, pre-eclampsia, and postpartum hemorrhage are significantly higher in women with intrauterine hematomas who deliver after 28 weeks of gestation, and these patients more commonly suffer from placenta previa and oligohydramnios than controls; therefore, this should be taken seriously by clinical workers [17].

#### 4.1.4. Autoimmune Disease

Autoimmune diseases seriously threaten maternal and infant safety by producing autoantibodies and immune complexes. It was proposed a long time ago that SCHs might be associated with autoantibodies, although this involved a small number of SCH cases (five with threatened abortion, comprising three with antinuclear antibodies and two with anticardiolipin antibodies), and the conclusion was that women with autoimmune diseases or antiphospholipid syndrome (APS) suffered pregnancy loss more often [21]. Therefore, it was suggested that patients with SCHs should be tested for autoantibodies regardless of their obstetric history. To examine the associative relationship among autoantibodies and intrauterine hematomas, a retrospective study recruited 54 women (8 with hematomas) with poor obstetric outcomes in 2003 and found that antiphospholipid antibodies (APLs) and antinuclear antibodies (ANAs) were present in 100% and 50% of cases, respectively. The study concluded that autoantibodies, especially APLs, may play a role in intrauterine hematoma development [22]. In a recent study, researchers recruited a total of 97 SCH patients and 130 control cases and found that a higher proportion of women in the SCH group had autoantibodies detected than in the control group (45.36% vs. 21.54%; *p* = 0.000). The positivity rates for ANAs (24.74% vs. 10.77%; *p* = 0.005) and APL (25.77% vs. 11.54%; *p* = 0.005) were significantly different between the two groups [23]. The study also concluded that the average birth weight was significantly lower for women with autoantibodies in the SCH group; however, there was no significant difference in terms of preterm delivery, cesarean section, or pregnancy complications, and most SCHs (96.25%) were absorbed before the twentieth gestational week.

#### 4.1.5. Blood System Abnormalities

Pregnancy can affect hematologic indices, either directly or indirectly, to induce different physiologic changes. Owing to the lack of well-controlled prospective studies to guide treatment decisions, there are significant challenges for hematology consultants. A few researchers reviewed four reported cases of antepartum management of a factor IX deficiency in the English literature, and reported the occurrence of a large, late-trimester SCH in a pregnant woman with a factor IX deficiency and with laboratory evidence of consumptive coagulopathy during treatment who was conservatively managed and had a successful outcome at term [24]. However, another study reported three cases with poor obstetric outcomes; one patient was homozygous for mutations in the methylene-tetrahydrofolate reductase gene C677T and suffered fetal demise at 30 gestational weeks, and the other two patients had a protein S deficiency and had second-trimester losses [25]. This finding suggested that SCHs may be associated with abnormal coagulative states and that the underlying etiology of SCHs may be the first indicator of potential thrombotic hemophilia. Townsley et al. shared a review specifically discussing the diagnosis and management of benign hematologic disorders occurring during pregnancy, which included anemia secondary to an iron deficiency, thrombocytopenia, inherited and acquired bleeding disorders, and venous thromboembolism (VTE) [26].

#### 4.1.6. Vaginal Dysbacteriosis

It was demonstrated a long time ago that a bacteriological infection was a possible cause of abortions before 16 weeks of pregnancy, which was derived from a prospective study that recruited 283 patients; 18 pregnancies (7.5%) ended in a spontaneous abortion, 44.4% of which were aborted and showed an infection, and 16.6% were diagnosed with an SCH [27]. A recent case-controlled study in 2012 included 47 pregnant patients with SCHs and 1075 patients as a control group to evaluate the vaginal flora, and revealed that the positive rates of coagulase-negative staphylococci (12.8% vs. 4.1%; *p* < 0.01) and *Gardnerella vaginalis* (12.8% vs. 2.5%; *p* < 0.001) were significantly higher whereas the negative rate of *Lactobacillus* was significantly higher than that of the controls (42.6% vs. 27.6%; *p* < 0.05). This suggested a possible association between SCHs in the first trimester and vaginal flora changes [8].

### 4.2. Relationship between SCHs and Pregnancy Outcomes

The impact of SCHs on pregnancy outcomes is controversial. A large SCH, which was first described by Breus in 1892 (Breus’ mole), is a serious condition that is frequently complicated by intrauterine growth retardation and intrauterine fetal death [28]. Most hematomas in the first trimester are small and associated with better pregnancy outcomes. A large number of studies on the clinical value of SCHs in the genesis of pregnancy complications has underscored the relevance of the considered clinical problem and the role of SCHs in the genesis of obstetric complications. Regarding the influence of first-trimester SCHs on pregnancy outcomes and the mode of delivery, a study by Janowicz–Grelewska et al. found that vaginal bleeding can be a prognostic factor for the mode of delivery, and a higher rate of cesarean sections was observed in patients with vaginal bleeding [20]. On the basis of a review of reports in the literature, data on the relationships between SCHs and other pregnancy outcomes were collected, as described below.

#### 4.2.1. Spontaneous Abortion

A first-trimester ultrasound finding of an SCH can be quite upsetting for a pregnant woman. Maso et al. analyzed 248 cases of pregnancies complicated by first-trimester intrauterine hematomas and found that clinical complications occurred in 38.5% of the cases (diverse outcome group). Spontaneous abortion (14.3%), fetal growth restriction (7.7%), and preterm delivery (6.6%) were the most frequent clinical conditions observed. In particular, intrauterine hematomas observed before 9 weeks significantly increased the risk of a spontaneous abortion (odds ratio 14.79; 95% confidence interval 1.95–112.09) [6]. Additionally, a retrospective study that included 185 pregnant women (119 women with SCHs) assessed whether a first-trimester SCH influenced the pregnancy outcome and whether vaginal bleeding was a prognostic factor for the pregnancy course, revealing that the “N” index (expressing the maximal length of the hematoma to the maximal length of the fetus) may be a useful predictor of the pregnancy course. They found that an “N” ratio equal to 2.5 or higher was associated with a risk of miscarriage and that a surface area of the SCH equal to 280 mm or greater was more likely to reveal vaginal bleeding [20]. The findings also suggested that there were no significant correlations between SCHs or vaginal bleeding and gestational complications such as premature birth, intrauterine growth retardation, gestational hypertension, abnormal volume of the amniotic fluid, parity and order of gestation, or delivery. In addition, other researchers concluded that intrauterine hematomas varied in size and location in the first trimester, but were mostly smaller than those that occurred in the second and third trimesters [15].

#### 4.2.2. Premature Birth

In a study that included 2172 women (17.9% with SCHs; mean largest diameter 2.1 ± 1.4 cm), researchers proposed that an SCH was not associated with any pregnancy outcomes at more than 20 weeks of gestation, including gestational age at delivery, preterm birth, birth weight, gestational hypertension, pre-eclampsia, placental abruption, intrauterine fetal death at more than 20 weeks of gestation, cesarean delivery, blood transfusion, or antepartum admissions [12]. In another study, investigators recruited 63,966 women (1.7% with SCHs) with a singleton pregnancy who underwent routine ultrasonography before 22 weeks at one institution and found that women with SCHs before 22 weeks of gestation were at an increased risk of placental abruption and preterm delivery, but were not at an increased risk of other adverse pregnancy outcomes [5].

However, other studies have emphasized the frequency of preterm labor in pregnant women with SCHs, and the presence of SCHs could lead to adverse pregnancy outcomes in the first trimester [6,20]. A retrospective cohort study of 22 cases of persistent SCHs with symptoms up to delivery among 4763 singleton pregnant women from 1985 to 1996 found that the onset of the symptoms showed two peaks, one at 9–11 and one at 30–31 weeks of gestation; 13.6% were spontaneous abortions, 77.3% were premature labor, 9.1% were full-term deliveries, and 17 cases needed tocolysis (which failed to prevent premature delivery in 16 cases) [29]. In addition, a cohort study compared 512 women (with an SCH on their first-trimester ultrasound) with 1024 women (without a first-trimester SCH), in which all patients underwent routine transvaginal cervical length measurements between 18 and 22 weeks, and found that the presence of an SCH was significantly associated with a shorter mean cervical length, particularly one below the tenth percentile (4.27 cm vs. 4.36 cm; *p* = 0.038 and 1.9% vs. 0.5%; *p* = 0.006, respectively). Preterm delivery was more common in the SCH group (12.5% vs. 7.3%; *p* = 0.001) [13]. This study concluded that a first-trimester SCH was related to a shorter cervical length as well as preterm birth, but that mechanisms other than cervical shortening may be involved in preterm birth among women with SCHs. In a recent report, the authors demonstrated an association with preterm birth, independent of the presence of symptoms of pelvic pain and/or vaginal bleeding, in a prospective observational cohort study that recruited patients with an intrauterine singleton pregnancy between 5 and 14 weeks of gestation between March 2014 and March 2016 [30].

#### 4.2.3. Low Birth Weight

According to the literature reviews, much work has examined the association between SCHs, especially SCHs with first-trimester bleeding, and the incidence of LBW [18,23]. A retrospective cohort study recruiting 353 women (65 of whom suffered from vaginal bleeding and 288 cases that did not) with two gestational sacs on the first-trimester ultrasound after the transfer of fresh embryos derived from autologous oocytes found that, except for the live-birth rates of women with first-trimester vaginal bleeding who had an increased prevalence of SCHs (26.2% vs. 1.7%), the live-birth rates of women with bleeding and with no bleeding were similar (87.7% vs. 91.7%) [31]. Another cohort study on IVF-ET or FETs also found that an SCH was associated with a lower birth weight in singleton pregnancies (3207.8 ± 595.7 g vs. 3349.2 ± 59.7 g; *p* = 0.03) [18].

## 5. Conclusions

An SCH is more common in first-trimester pregnancies, but causes fewer complications in pregnancy. Although an SCH is less common in the second and third trimesters, it easily causes more pregnancy complications. Our research explored the lack of an effective predictive value between premature birth and the coexistence of SCHs with HDCP, IVF, or PROM in the third trimester; the latter increased the rate of cesarean sections through inducing the anxiety of pregnant women and their families. Therefore, this finding provides evidence to help clinicians comfort patient and relieve their anxiety, thus reducing the cesarean section rates. To better understand the effect of SCHs on adverse pregnancy outcomes, large-scale and multicenter trials taking a detailed clinical index associated with SCHs into consideration are needed when we evaluate their significance in the prognosis of pregnancies in the future.

## Figures and Tables

**Table 1 jpm-13-00479-t001:** Descriptions of clinical features analyzed in this study as related factors with the pregnancy-related diseases of patients with SCH.

Variable		All Per (Cases)	GDM			HDCP		
			Per (Cases)	χ^2^	*p*-Value ^a^	Per (Cases)	χ^2^	*p*-Value ^a^
Gravida	Primi	49.1% (546)	35.6% (48)	11.280	0.001	56.8% (46)	2.067	0.151
	Multi	50.9% (566)	64.4% (87)			43.2% (35)		
Parity	Primi	75.1% (835)	77.0% (104)	0.311	0.577	79.0% (64)	0.719	0.397
	Multi	24.9% (277)	23.0% (31)			21.0% (17)		
Gestation	Mature	89.0% (990)	87.4% (118)	0.414	0.520	76.5% (62)	13.943	0.000
	Premature	11.0% (122)	12.6% (17)			23.5% (19)		
Age	Normal	82.6% (919)	69.6% (94)	18.143	0.000	70.4% (57)	9.175	0.002
	Advanced	17.4% (193)	30.4% (41)			29.6% (24)		
Delivery	Vaginal	57.5% (634)	39.3% (53)	19.764	0.000	37.0% (30)	14.227	0.000
	Cesarean	43.0% (478)	60.7% (82)			63.0% (51)		
**Variable**		**All Per (cases)**	**IVF**			**PROM**		
			**Per (cases)**	**χ^2^**	***p*-Value ^a^**	**Per (cases)**	**χ^2^**	***p*-Value ^a^**
Gravida	Primi	49.1% (546)	45.6% (52)	0.618	0.432	53.5% (83)	1.426	0.232
	Multi	50.9% (566)	54.4% (62)			46.5% (72)		
Parity	Primi	75.1% (835)	92.1% (105)	19.661	0.000	77.4% (120)	0.522	0.470
	Multi	24.9% (277)	7.9% (9)			22.6% (35)		
Gestation	Mature	89.0% (990)	81.6% (93)	7.217	0.007	81.9% (127)	9.278	0.002
	Premature	11.0% (122)	18.4% (21)			18.1% (28)		
Age	Normal	82.6% (919)	68.4% (78)	17.914	0.000	82.6% (128)	0.001	0.982
	Advanced	17.4% (193)	31.6% (36)			17.4% (27)		
Delivery	Vaginal	57.5% (634)	25.4% (29)	51.675	0.000	65.2% (101)	4.878	0.027
	Cesarean	43.0% (478)	74.6% (85)			34.8% (54)		

^a^ Chi-squared test or Fisher’s exact test were performed to compare the two groups. Per: percentage.

**Table 2 jpm-13-00479-t002:** Analysis of the associations between premature birth and advanced age with pregnancy-related diseases in SCH patients.

Variable		Gestation		*p*-Value ^c^	Age		
		Mature	Premature		Normal	Older	
		Mean ± SD (Weeks)			Mean ± SD (Years)		*p*-Value ^c^
		39.16 ± 1.09	34.35 ± 2.35		29.85 ± 2.67	37.00 ± 2.31	
GDM	Yes	118	17	0.240	94	41	0.001
	No	872	105		825	152	
		38.89 ± 1.07	34.26 ± 2.21		29.82 ± 2.75	37.21 ± 3.28	
HDCP	Yes	62	19	0.002	57	24	0.862
	No	928	103		862	169	
		38.95 ± 1.07	34.67 ± 2.18		31.12 ± 2.35	37.78 ± 3.27	
IVF	Yes	93	21	0.009	78	36	0.168
	No	897	101		841	157	
		38.96 ± 0.98	34.16 ± 2.52		29.83 ± 2.53	36.59 ± 1.80	
PROM	Yes	127	28	0.000	128	27	0.150
	No	863	94		791	166	

^c^ T-test was performed to compare the two groups. SD: Standard Deviation. GDM: gestational diabetes mellitus, HDCP: hypertensive disorder complicating pregnancy, PROM: premature rupture of membranes, IVF: in vitro fertilization.

**Table 3 jpm-13-00479-t003:** Effects of different gravidity and parity on gestational weeks of SCH patients.

Variable		N	Gestation	F	*p*-Value ^b^
			Mean ± SD (95% CI)		
Gravida	1 = primi_G	291	38.10 ± 2.20 (37.85–38.35)	4.965	0.026
	2 = multi_G	467	37.72 ± 2.31 (37.51–37.93)		
Parity	1 = primi_P	513	38.07 ± 2.21 (37.87–38.26)	12.236	0.000
	2 = multi_P	245	37.45 ± 2.36 (37.16–37.75)		

^b^ One-way ANOVA was performed for a comparison among groups. SD: Standard Deviation. CI: confidence interval. F: F-test.

## Data Availability

All data in this study derived from medical record information base in Shanghai First Maternal and Infant Health Hospital, School of Medicine, Tongji University, Shanghai, China.

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
