# Peer review of "Subchorionic Hematoma Association with Pregnancy Complications and Outcomes in the Third Trimester"

_jpm, 2023, doi:10.3390/jpm13030479_

Round 1

Reviewer 1 Report

According to the manuscript submitted, there are several comments and inquiries.

1. What is the primary objective of this study? It was so confusing that the aim of this study was to evaluate the association of subchorionic hematoma in the third trimester of pregnancy and the Obstetric complications or to evaluate the association of subchorionic hematoma and the Obstetric complications in the third trimester.

2. What was the reason to conduct this study? 

3. Please describe the definition of subchorionic hematoma, the diagnostic criteria of GDM and HDCP.

4. Please provide the details of sonographers who evaluated SCH in this study and also the reliability of the report.

5. Were those with spontaneously resolved SCH included in this study?

6. The results of this study could not be concluded as the sample size was accurately calculated.

7. The tables of the results were so confusing. Please rearrange and summarized the important issues.

8. In the discussion part, the authors should discuss the results and compared to previous studies. There's no need to describe the details of previous studies.

Author Response

Dear editors and reviewers,

Thank you very much for your comments on our manuscript entitled “Subchorionic Hematoma Association with Pregnancy Complications and Outcomes in the Third Trimester” (JPM, ISSN 2075-4426). Please kindly accept our apologies that we have not respond you timely. We have read your comments very carefully and decide to resubmit the manuscript. Detailed supplementing related information is as follows:

  1. What is the primary objective of this study?It was so confusing that the aim of this study was to evaluate the association of subchorionic hematoma in the third trimester of pregnancy and the Obstetric complications or to evaluate the association of subchorionic hematoma and the Obstetric complications in the third trimester.

The occurrence of subchorionic hematoma greatly leaded to the anxiety of pregnant women and their families, and then which induced some obstetric related complications and the higher cesarean section rate. Our primary objective was to explore the complications and outcomes of subchorionic hematoma (SCH) in the third trimester, and found the coexistence of SCH with HDCP, IVF, or PROM lacked effective predictive value for premature birth but increased the rate of cesarean section. Therefore, it can help clinicians comfort and relieve the anxiety of pregnant women and reduce the cesarean section rate.

  1. What was the reason to conduct this study? 

As explained above, we conduct this study to confirm the lacked effective predictive value between subchorionic hematoma in the third trimester and the obstetric complications, and then comfort and relieve the anxiety of pregnant women and reduce the cesarean section rate.

  1. Please describe the definition of subchorionic hematoma, the diagnostic criteria of GDM and HDCP.

We have defined the subchorionic hematoma in the first sentence as “A subchorionic hematoma (SCH) is defined as a crescent-shaped, echo-free area between the chorionic membrane and the myometrium”, and supplemented the information of the diagnostic criteria of GDM and HDCP in the manuscript marked up using the “Track Changes” function.

  1. Please provide the details of sonographers who evaluated SCH in this study and also the reliability of the report.

The relevant information has added in Materials and Methods.

  1. Were those with spontaneously resolved SCH included in this study?

Yes, SCH was the common sonographic abnormality during pregnancy and the spontaneously one was included in our study.

  1. The results of this study could not be concluded as the sample size was accurately calculated.

Our calculation was gone through serious statistical analysis and then came out the conclusion.

  1. The tables of the results were so confusing. Please rearrange and summarized the important issues.

We have made the table again according to the three-line method.

  1. In the discussion part, the authors should discuss the results and compared to previous studies. There's no need to describe the details of previous studies.

Thanks for this valuable opinion and suggestion, we aim to make the article more lively and to illustrate a more objective conclusion through literature elaboration and comparison.

Reviewer 2 Report

The article is valuable because it presents statistics on the disease. Because it presents the data of the disease from different angles, a different perspective is presented.  That's why the article is important. 

Author Response

Dear editors and reviewers,

Thank you very much for your recognition and support. We wish to be considered for publication in the Journal of Personalized Medicine and hope a good luck!

Reviewer 3 Report

I found this study makes a valuable contribution to literature and the field.

1.       The discussion section is too big and confusing; authors must focus on their significant findings and discuss them with exciting literature.

2.       The conclusion must be rewritten to show the importance of the findings to the research question. 

Author Response

Dear editors and reviewers,

Thank you very much for your comments on our manuscript. Please kindly accept our apologies for consuming your time to revise our article again.

  1. The discussion section is too big and confusing; authors must focus on their significant findings and discuss them with exciting literature.

Thanks for this valuable opinion and suggestion, we're not just trying to elaborate on previous literature reports but also aim to make the article livelier and to illustrate a more objective conclusion through literature elaboration and comparison.

  1. The conclusion must be rewritten to show the importance of the findings to the research question.

The relevant information has been rewritten and added in Conclusion after our serious modification. The information is as follows:SCH is more common in first-trimester pregnancies but causes fewer complications in pregnancy. Although SCH is less common in the second and third trimesters, it easily causes more pregnancy complications. Our research explored the lacked effective predictive value between premature birth and the coexistence of SCH with HDCP, IVF, or PROM in the third trimester, but the later has increased the rate of cesarean section through inducing the anxiety of pregnant women and their families. Therefore, this finding aimed to provide evidence to help clinicians comfort and relieve the patients’ anxiety and then reduce the cesarean section rate. To better understand the effect of SCH on adverse pregnancy outcomes, large-scale and multicenter trials taking a detailed clinical index associated with SCH into consideration are needed when we evaluate its significance in the prognosis of pregnancy in the future.

Round 2

Reviewer 1 Report

The revised version was well-written and the results of this study would be useful for patient counseling.